# Regulation and Certification of (Bio)Medical Engineers: A Case Study on Romania

**DOI:** 10.3390/ijerph19159004

**Published:** 2022-07-24

**Authors:** Marian Miculescu, Oana Andreea Ion

**Affiliations:** 1Department of Metallic Materials Science and Physical Metallurgy, Materials Science and Engineering Faculty, University Politehnica of Bucharest, 060042 Bucharest, Romania; 2Department of International Relations and European Integration, National University of Political Studies and Public Administration, 012104 Bucharest, Romania; oana.andreea.ion@dri.snspa.ro

**Keywords:** biomedical engineering, education, profession, qualification, Romania

## Abstract

This paper analyzes the Romanian biomedical engineering educational path and certification process in European and international contexts and emphasizes the existence of a deficient operationalization of this qualification and profession, arguing that the domestic shortcomings are both a consequence of an unquestioned process of adopting European and even international classification schemes, and of insufficiently developed national administrative capabilities to properly implement the aforementioned classification frameworks. The core part of the article investigates the current academic track of the biomedical engineering specialization and scrutinizes the classification of occupations at different jurisdictional levels. The conclusions of the study indicate that one of the possible solutions for improving this unsatisfying *status quo* comes from a better communication between the national and European levels, and by their pro-active involvement in the international attempts of reviewing and refining the existing frameworks. The article ends with several recommendations and policy proposals meant to strengthen the role of various profession-certifying European documents, as well as to alleviate the regulatory deficiencies that this specialization has at Romanian level, in order to maximize its potential in the labor market.

## 1. Introduction

Getting your professional qualifications recognized worldwide is still a goal too far away to be obtained even in today’s globalized economy, as revealed by the drafting and ratifying process of UNESCO’s Global Convention on the Recognition of Qualifications concerning Higher Education [1]. Nevertheless, there are some exceptions coming from regional international organizations that understand the need to cooperate in various fields in order to provide their citizens real political, economic and social opportunities in the current competitive environment. One example—rather singular, in fact—is the case of the European Union and its permanent quest for improving the *acquis* on professional recognition (at least) among its member states. However, due to the intricate balance between EU and national competencies in policies regarding education, employment, health, social benefits, etc., there is not a unified EU system of recognizing academic diplomas and, therefore, professional qualifications; these procedures are being tackled at the national level. 

This article belongs to the general literature on the unsatisfactory situations associated with qualifications and professional recognition and its purpose is to determine the effects of the EU member states’ mimetic adhesion to European and/or international various standards in the mentioned domain, and to propose a set of policy initiatives meant to alleviate the existing national inadequacies. What are the causes of the poor operationalization of some qualifications and professions, both at the member states’ level and at the broader EU level? In other words, are some existing domestic deficiencies in the operationalization of some professions a consequence of an unquestioned process of adopting European and even international classification schemes, or can these shortcomings be explained by poor national administrative capabilities of properly implementing the aforementioned schemes? This research question will be answered by investigating the case study of the biomedical engineering (BME) domain by a triple perspective: international, European and national, where we have selected the Romanian example. The gathering of data process implied primary sources under the form of official inter/governmental documents; considering the mixed character of our research question, for the numerical analyses of the collected data and for the meaning interpretation of their content, we used a combined quantitative and qualitative methodology. 

There are two main parts of the article: firstly, we explain the research problem and we offer a non-classical description of the biomedical engineering profession, addressing questions about the role of the biomedical engineers in today’s and tomorrow’s society and analyzing the improper definitions of the BME field that negatively impacted its development. Secondly, we investigate the current academic track of the BME specialization, pointing towards the diversity of educational perspectives that are supposed to lead to the same or, at least, compatible learning outcomes. Then, we scrutinize the classification of occupations at different jurisdictional levels and we emphasize the similarities and discrepancies in the BME case. The conclusions of the study indicate that one of the possible solutions for improving this unsatisfying *status quo* comes from a better communication between the national and European levels, and by their pro-active involvement in the international attempts of reviewing and refining the existing frameworks. In this respect, we address several recommendations pointing toward a stronger role of various European instruments, such as the European Qualifications Framework, the regulated professions database or the European Skills, Competences, Qualifications and Occupations codification. For the national level, we elaborate on a policy proposal of modifying the Romanian Classification of Occupations Register in order to ensure a better compliance between the BME field and the European and international frames. In addition, based on the BME professionals’ fundamental societal input, we conclude by elaborating a recommendation meant to increase their national employability.

## 2. Materials and Methods

In light of the previous arguments, the main research objective of the article is to identify several causes of the inadequate operationalization of some qualifications and professions at national and EU level, with a study case focused on the BME field in the Romanian context. The results will be relevant for both the domestic and European academic community, considering, on the one hand, its novelty, as the research covers a subject insufficiently exploited in the literature, and, on the other, that it will also propose several policy initiatives meant to alleviate the existing international, European and national inadequacies. With a focus on the domestic level, where the majority of the shortcomings occur, the conclusions underline the need for a more active role for decision-makers in shaping national, European and international frameworks, because the supra-national frameworks contain provisions that can only be modified with the cooperation of the state actors. 

Starting from the above mentioned research questions, the first research hypothesis of this article is that the existing domestic deficiencies in the operationalization of some professions are a consequence of an unquestioned process of adopting European and even international classification schemes. The second research hypothesis argues that these shortcomings can be explained by poor national administrative capabilities of properly implementing the aforementioned schemes. 

In order to validate these hypotheses, data about the BME domain were gathered, following a tripartite structure: international, European and national, where we have selected the Romanian study case. Because our research has a descriptive and explanatory purpose, we used both quantitative and qualitative research methods. For the description of the BME profession and for assessing its current societal role, we used, as qualitative method, the content analysis instrument, investigating various official documents provided by international actors (either political, such as the European Commission, or sectoral/professional, such as World Health Organization or European Alliance for Medical and Biological Engineering & Science) or articles from the few academic literature developed on the subject. The analysis of the educational and of the professional track of the BME graduates combines the qualitative and quantitative approach, being focused mainly on official sources provided by international, European and national authorities. The description of the BME codifications in various classification schemes, as well as the latest data provided by the Romanian institutional bodies (professional regulations, as well as the sampling of the BME fields and programs of accredited university and master studies), were investigated and interpreted both qualitatively (content analysis) and quantitatively (numerical analysis). 

In the continuation of this section, we provide an in-depth insight of the BME professional and education institutionalization in the context of a prominent and increasing societal role that does not benefit, however, from a harmonized perception on its definition and implications. Our paper uses extensively primary sources under the form of official documents issued by international, European and national authorities because, after an extensive desk research of the existing literature, we noticed that there are very few similar and even fewer recent contributions published in the last ten or, more specifically, the last five years. Some focus only on the situation of BME programs in Europe in order to identify development trends [2] or only on the global situation of BME undergraduate programs [3], whereas others [4] concentrate on another case study about the need to certify the “clinical engineering” branch. There are also some contributions focused on the personal development of BME graduates [5] or on promoting the inclusion and diversity of candidates and, implicitly, graduates [6]. In connection with the subject of our paper, there are no references to other similar cases in national or wider European contexts that would have been more or less successfully conceptualized and implemented, serving or not as best practice examples. An explanation for this scarcity of data from scientific papers from recent years may come from the status of education and health policies (this article being at their confluence), where both problems and solutions are addressed mainly at the national level. However, this possible limit of our paper represents, at the same time, an advantage, transforming it in an innovative analysis of primary data retrieved from official documents.

### 2.1. Institutionalizing Biomedical Engineering

The BME field should not be seen as a complete novelty regarding both professional and educational aspects. We do not refer here to the fact that various medical devices meant to improve the life of patients could be found even since the ancient Egypt [7], but to (a) the post-WWII innovations that started to connect engineering and medicine for improving the quality of services in medical and healthcare systems; (b) the international BME professional organizations established since the late 1950s (IFMBE, for example, was established in 1959); and (c) the first graduate and undergraduate-accredited BME programs flourishing in the United States since the early 1960s and, respectively, 1970s, the latter as a consequence of the universities’ intentions to fully dispose from their financial resources and to elaborate on a curriculum without external interference. Without a concrete labor-market-specific demand for BME professionals (as the tasks were solved with practitioners from various engineering occupations), the field was therefore developed in universities, where the engineering specialists opened towards the challenges posed by medicine and biology. In this case, the supply modeled the demand. Since the 1970s, the BME in the US developed constantly in the number of programs (118 undergraduate programs in 2018), of BME-enrolled students or of academic staff [3,8], while careers as bioengineers and biomedical engineers are well-positioned in the US society [9]. 

However, if the US figures sustain the image of the BME field as a fully entitled type of engineering specialization, the European experience is different. The US case, illustrative to present the emergence and development of BME, cannot be used for a solid comparative analysis with the European one due to too large differences between educational systems and career opportunities. If one considers, as a minor example, that even the credits of a course are different in the US [7,10] and in Europe (which has almost entirely adopted the ECTS system), one can understand at another level the complexity of a comparative approach. Given Romania’s status as a member of the EU, we consider that, for our analysis, it is more relevant, after a more general discussion about the BME status at international level, to focus on the case of Europe, and to finally reach an analysis of the national level.

It is argued that the ideal-type institutional development of a profession should contain several landmarks: establishing training and, later, higher education institutions; setting up local level and national level professional associations; developing certification regulations, as well as a proper code of ethics [11] (p. 59). In this article, without denying the role of professional associations or codes of ethics, we will focus on the educational dimension (especially the higher education aspect) and the certification process implied by the need of professional recognition. 

A follow-up would then be necessary to complete the image on the institutionalizing of the biomedical engineering realm. Future analyses should be extended, for example, towards the profile and role that can be played in better defining and supporting the development of the BME specialization and workforce of professional associations, such as International Union for Physical and Engineering Sciences in Medicine (IUPESM), International Federation for Medical and Biological Engineering (IFMBE), Australasian College of Physical Scientists and Engineers in Medicine (ACPSEM), Consejo Regional de Ingeniería Biomédica para América Latina (CORAL), European Alliance for Medical and Biological Engineering and Science (EAMBES) and European Society of Engineering and Medicine (ESEM), to name but a few of the international or regional existing bodies, as revealed by the WHO data [11] (p. 57). In Romania, the same WHO document indicated for 2016 the existence of a single national professional biomedical engineering association, namely “Societatea nationala de inginerie medicala si tehnologie biologica” (SNIMTB—National Society of Medical Engineering and Biological Technology). Our current research indicates that SNIMTB is affiliated to IFMBE, and that its members include few persons from the HE institutions that offer BME specializations in Romania. Moreover, there is no evidence of a code of ethics on the nongovernmental organization’s website.

### 2.2. BME’s Societal Role

Faced by an aging population, rise of chronic diseases and low birth rates, the future of the European population is threatened not only by insecurities connected to pensions payment, but also to health and care services that are at the same time accessible, affordable and also respecting safety and quality criteria in a both efficient and sustainable manner. European citizens will soon need an extended ability to retire later and to benefit from the outputs of personalized medical services and devices. In 2015, the European Economic and Social Committee adopted a document recommending joint action of the biomedical engineering domain with the medical and care services industry, arguing that modern medicine cannot be conceived without the biomedical engineering industry, irrespective of if the final aim is the general improvement of the quality of life or the more focused one of enhancing the healthcare systems. However, the most comprehensive picture of the BME societal role is to be found in the records of the European Alliance for Medical and Biological Engineering and Science (EAMBES), a professional association describing its connections with:
“every industrial sector developing products where technologies of any kind interact with the human body: medical devices (cardiac valves, hip replacements, etc.), medical technology (Computed Tomography scanners, electrocardiograms, etc.), Sport, fitness and wellness equipments (running shoes, training machines, etc.), defence & security (body armours, security scanners, etc.), ergonomics and safety (ergonomic tools, car airbags, etc.), entertainment (motion capture for computer graphic animations and games, etc.). In the future this might also include applications that are currently at the research stage, such as brain-computer interfaces, wearable or implanted technologies (i.e., implanted tags in prison inmates), etc.”.[12]

The most recent World Health Organization (WHO) data indicate that BME professionals were present in 64% of the WHO member states, with a clear growth tendency, considering, on the one hand, the low and medium income countries’ interest for educating and training for this qualification and, on the other, the desirability of the profession according to a 2012 United States Bureau of Labor Statistics survey that included criteria such as the employability, work conditions, income, levels of physical demand and stress [11] (pp. 26, 37). The US can be used here as a benchmark; the BME sector is a job market that has developed rapidly in the last decade, but which currently has a development trend close to the media of other specializations [9], although, in terms of salaries, they are almost double the median annual salary for total US occupations.

In Europe, seen also as the engineering discipline with the most accelerated development and some of the most attractive career trajectories, BME covers a market size of EUR 140 billion (the second medical technology market in the world, with the highest number of patents), with over 33,000 companies and over 750,000 (usually high skilled) employees, benefitting from large amounts from the research funds addressed to health programs during the latest multi-annual financial frameworks [7,13,14]. Nevertheless, the European landscape is rather fragmented in comparison to the US experience due to the lack of accommodation between the EU and national level visions on a BME sector that still lacks the independent methodological and analytical character that is benefited from in the American version [15]. For example, the EU’s research expenditures on health programs (in)directly connected with (bio)medical engineering did increase over the years, following the general trend of EU funds allocated to research [14,16,17], but a proper identification of the specific health amount is still difficult, considering that this is not a common EU policy. Therefore, funds for R&D in medical and care sectors are diluted among various EU programs and policy priorities, and later under the Horizon 2020 chapter.

Is this fragmentation a singular mark of the European experience or does it have more complex causes that can be explained in a larger international context?

In order to answer this question, we used the data provided by one of the most involved international actors in the survey of the BME realm, which is WHO, and which has published, in the last decade, in-depth related analyses on countries, professionals, educational and training institutions, professional societies, etc.

Despite a narrow focus on the medical devices case, the report “Human Resources for Medical Devices. The role of biomedical engineers” issued in 2017 (and based on a wide WHO research program entitled “Biomedical engineering global resources”) reveals, as one of the most comprehensive perspectives on the definition, formation, employment and perspectives of development for the biomedical engineering sector, seen as an essential part in the process of translating health policies in efficient and sustainable health outcomes, as presented in Figure 1:

This focus on biomedical engineering is motivated by the wide array of activities that it supports by producing knowledge and by fostering the application of new medical devices and services. From a larger perspective, BME specialists should also be able to coin solutions for problems that do not exist yet. Compiling definitions and role descriptions offered by various stakeholder institutions, one broadly refers to:
“designing, manufacturing, marketing, procurement, regulating, evaluating, daily operations of monitoring, managing maintenance and repairs, as well as training for (safe and effective) use, of medical devices and equipment, special (intelligent) materials, implantable devices, artificial organs, prostheses and robotic systems for biomedical applications, including information systems and software for processing biomedical and bio-imaging data. Within this prevention/diagnosis/treatment/care and rehabilitation logic, it also includes ICT focus, biotechnology and cell engineering, nanotechnology, modelling and simulation of physiological systems and the human body as a whole, or development of minimally invasive surgical techniques, as well as development of medical devices policies”.(compilation from [11,15,18])

Therefore, evidence indicates that the BME role is rather complex. Is this agreement regarding the societal importance of BME also transferred in a unitary definition of the field?

### 2.3. BME: A Common Understanding of the Discipline?

From the activities and roles presented above, what can be briefly argued about the profile of a biomedical engineer? One definition is supported by a professional association and stipulates that “medical and biological engineering integrates physical, mathematical and life sciences with engineering principles for the study of biology, medicine and health systems and for the application of technology to improve health and quality of life” [18]. Following the same logic and stressing the interdisciplinary dimension, “biology, physics, medicine, engineering, nanotechnologies, and ICTs” are encouraged to work together “in order to address the important challenges in healthcare and market opportunities” [14]; or, “[b]iomedical engineering is an interdisciplinary field in which all engineering and technological sciences are integrated in solving the problems that arise in the field of biology and medicine” [19]. In other words, but the same idea: “applying knowledge of engineering and technology to health-care systems to optimize and promote safer, higher quality, effective, affordable, accessible, appropriate, available, and socially acceptable technology to populations” [11] (p. 23). A slightly different note can be found here: “Biomedical engineering is a cross-disciplinary science based on medicine, biology and engineering” [15] (p. 5), while “a necessary condition for future progress in this exciting cross-disciplinary area depends on improved communication and synergy between all the research disciplines and their concerted effort with industry and relevant authorities” [14].

We argue that one of the problems regarding the BME status comes from a poor operationalization of the terms used for describing the stakeholders’ position on the matter. There are considerable reasons to favor a multiple disciplinary approach: to solve current and complex challenges, to bring new viewpoints on a problem in order to maximize the number of possible solutions, to strengthen the research potential by enlarging the theoretical base that can generate testable hypotheses, etc. [20]. However, is BME cross-disciplinary, multi-disciplinary, inter-disciplinary, trans-disciplinary or does it not matter so much, as all of these concepts can be considered synonymous?

Lexical semantics studies underline the existence of significant differences between the meaning of the afore-mentioned concepts, all of them connected to the pluralistic disciplinarity approach logic. One could envisage them, like in Table 1, on a continuum from a mere juxtaposition of various disciplines to a complex fusion:

In our view, BME is a field that currently stands under the inter-disciplinary label and it would be improper to perceive it as a mere cross or multi-disciplinary field. The most succinct definition, “Medical and Biological Engineering Sciences is an autonomous scientific research discipline, which generate new knowledge on biological systems using the methods and the approaches that are proper of physical and engineering sciences”, presented on the EAMBES [12] website, is, paradoxically, useful; avoiding fashionable terms such as *integration*, *principles*, *working together*, *communication*, *synergy* or *applying knowledge*, EAMBES bluntly states that “the development of biomedical engineering sciences (…) involves the generation of new knowledge on biological systems using the methods and the approaches that are proper of physical and engineering sciences”. Therefore, a new integrated research area operating under an inter-disciplinary logic. 

The discussion on inter-disciplinarity is extremely relevant for the next section dedicated to the BME educational path and professional recognition, but especially for formulating our concluding recommendations, because we consider that a closer analysis of the subject would optimize the adequate institutionalization of BME in time and quality.

## 3. Results and Discussion

### 3.1. BME: Educational Track

So far, we have used the term biomedical engineer, as it is the most common and inclusive concept present in the literature. However, the aforementioned large array of activities that a biomedical engineer could be involved in is illustrative of an internal diversity of the possible job sectors, positions and field descriptions, as well as of various denominations used interchangeably: bioengineer, medical engineer, clinical engineer, biological engineer, etc. These various profiles reflect the complexity of this science domain that already has its own distinctive sub-domains [7], but are gathered under the BME general label. Only a part of this variation can be explained via the educational track—a rather unitary core curriculum foundation, followed by subsequent specializations. 

There is an impressive educational offer on BME specializations at the general global level and, in aspects concerning our analysis, in *Europe*’s case, as indicated in a WHO survey. The majority of the investigated programs (over 300 case studies) focus on disciplines connected with engineering, physical sciences, biology and medicine, which is consistent with worldwide best practices, where the core curriculum includes human anatomy and physiology, as well as engineering, while elective specialization implies disciplines such as artificial organs and support systems, biomaterials, clinical engineering, computational modeling, implants and prosthetics, neural engineering, regulatory standards, rehabilitation, process and systems engineering, etc. [11] (pp. 39–47).

WHO 2009–2015 investigated data also indicated that, in Europe, BME specializations were offered for both undergraduate (first cycle, BSc) and graduate (second cycle, MSc; the PhD courses were rare) levels, with a predominant presence of the MSc programs having in mind the usual perception that a “biomedical engineer is […] an engineer cross-trained and specialized in biomedical application areas” [11] (p. 46). The same type of data provided in 2021 by EAMBES through INBIT, a non-profit BME oriented organization, indicated the presence of 117 B.Sc. programs, with, comparatively, 167 M.Sc. programs (confirming the second cycle stronger input), but also the presence of 56 PhD programs, a context that indicates a progressive interest towards this field of studies [24]. We have referred above to the usual perception of a biomedical engineer because the entry rules for some of these second--cycle programs allow students to come not only from BSc programs from the BME or engineering/physical areas but also from biological or medical programs. Moreover, the whole picture is complicated by the current 3+2 Bologna system, which was not successfully implemented all over the engineering specializations, including BME, a fact that generates graduates from integrated first and second-cycle programs [11] (pp. 46–47). 

In *Romania*, the WHO survey indicated only two HE institutions offering biomedical engineering programs: “Grigore T. Popa” University of Medicine and Pharmacy from Iasi (unspecified program level) and University “Politehnica” of Bucharest (MSc level—Bioinformatics/Medical and Clinical Engineering/Biomateriale). Nevertheless, we will not focus on these data, as there are several signs of doubt regarding their accuracy. First, because University “Politehnica” of Bucharest also offers two B.Sc. programs, the first one (in Eastern Europe) being operational since early 2000 and the other since 2010 [25,26], situation not reflected in WHO’s survey. Second, additional WHO data about “Biomedical engineering professionals per 10,000 population in the WHO European Region” or about “Reported density of hospitals with biomedical department/unit/service per 100,000 country population by WHO region” [11] (pp. 34–35) seem to reflect a favorable image of Romania (placed 12th from 40 investigated countries, in the first case, and placed 7th from 12 countries, in the second case, with notorious absences). Nevertheless, without proper details about the methodology used to provide these statistics and with an inside vision on the Romanian shortage of BME professionals and BME services provided in hospitals [27], these data do not offer a realistic image on the Romanian BME educational and professional field. The more recent EAMBES database (accessible through INBIT 2021 [24]) should also be tacked with prudence, considering that Romania is present with only one program for each level (B.Sc. and M.Sc.), whereas the official governmental documents reveal a different picture, as indicated below.

The educational picture of the BME specialization in Romania is detailed for the 2021–2022 academic year by two governmental decisions (see [28,29]). As indicated in Table 2, as part of the broader “Engineering Sciences” educational area, there are two specializations (medical engineering and, respectively, bioengineering) that belong to the B.Sc. and M.Sc. field of “applied engineering sciences” within the specific branch of “mechanical engineering, mechatronics, industrial engineering and management”.

The same legal documents indicate which higher education institutions (HEI) offer BME programs for both B.Sc. and M.Sc. levels, also pointing out the status of the program (either accredited or just under a provisional operating authorization) and the maximum number of students allowed to be enrolled on budgeted places. For a more fluent vision of the data, we have eliminated several indicators that did not affect our analyses (the number of credits—240 for B.Sc. and 120 for M.Sc.; the location of the program—with one exception, all are offered within the towns where the HEI is located; or the mention that they are all full-time programs).

For the B.Sc. level, one can observe in Table 3 that there are eight HEI that offer nine BME programs (even if one of them did not enrol any student in the mentioned academic year). Among these programs, eight are for the *medical engineering* specialization and one is for the *bioengineering* specialization.

For the M.Sc. level, one can observe in Table 4 that there is only one HEI offering a BME program. 

We conclude this section by arguing that, from the above tables, one can see that Romania offers the educational framework for specialists in two occupations associated with the BME sector—medical engineers and bioengineers, both with the same number of transferable credits for the first cycle of studies, so with the same workload. Hence, whereas the bioengineering specialization is followed only at the level of the B.Sc. of a single HEI, the medical engineering is offered by eight HEI (three having the status of PA, so showing the new and growing interest in the field). In addition, medical engineering is the only specialization also offered at the M.Sc. level.

### 3.2. Professional Recognition

We argued earlier that only a part of the various BME-related job titles and fields can be explained via the educational track. The general explanation, in our view, lies within the unsettled inter/national legislation that normatively codifies this profession.

At international level, ILO—through its ISCO 08: International Standard Classification of Occupations—does not have a specific place for biomedical engineers in the minor group of the engineering professionals, and they are included in the general unit group “Engineering Professionals Not Elsewhere Classified”, with a particular emphasis stating that “while they are appropriately classified in this unit group with other engineering professionals, biomedical engineers are considered to be an integral part of the health workforce alongside those occupations classified in Sub-major Group 22: Health Professionals, and others classified in a number of other unit groups in Major Group 2: Professionals” [30] (p. 120). This disclaimer is extremely important for the confusion that sometimes accompanies the BME placement in the classification of occupations, and it emphasizes the connection between the BME profession and the medical and health-care workforce, supporting national and international approaches that underline the fundamental role BME plays for the health systems, without (n.a.) asking for classifying BME under the medical workforce. 

At the European Union level, the European Skills, Competences, Qualifications and Occupations (ESCO) description claims that it “identifies and categorises skills, competences, qualifications and occupations relevant for the EU labor market and education and training” [31], but the adequacy of this statement is debatable, considering that it actually duplicated the ILO system at the EU level with a single—important—trump card, the legal basis of enforcing this uniform classification within the member states. The ESCO website offers additional information about the alternative labels used for this occupation, as well as about the required essential and optional skills and competences. However, for the aims of this article, there are three other points that draw our attention. Firstly, the description of the occupation: “Bioengineers combine state of the art findings in the field of biology with engineering logics in order to develop solutions aimed at improving the well-being of society. They can develop improvement systems for natural resource conservation, agriculture, food production, genetic modification, and economic use”. Secondly, the so-called essential knowledge; in other words, the formative fundamental disciplines identified as biological chemistry, biology, engineering principles, engineering processes and genetics. Thirdly, the invitation to consult the Regulated Professions Database of the Commission in order to see how this occupation is regulated in the larger European economic area. However, this encouragement for the search of new information is just a usual disclaimer found on any EBSO occupation description page, as the mentioned database, connected to the free movement of professionals EU concern [32], does not contain any evidence of the bioengineer item, as the only relatively connected occupation present in the register is medical/biomedical laboratory technician [33]. In fact, this indicates that BME cannot be considered as a regulated profession within the EU member states, as “here is no centralized, common certification program that establishes certification standards” applicable at the Union’s level [11] (p. 68).

At the national level, the structure of the classification of occupations in Romania (COR), as indicated by the Romanian Ministry of Labour website, is a consequence of the EU legislation interested in providing “comparability between data on occupations from the EU Member States and the rest of the world” [34] and arguing in favor of the national implementation of the International Standard Classification of Occupations revised classification (ISCO-08). However, the process of recodifying did not only imply that several groups and occupations were taken for granted when translated into the new system; in Romania, this process also meant that, in some cases, without any explanation, some occupations were totally omitted or placed in a different place, as in the BME case. 

Table 5 indicates that the biomedical engineer occupation (with its alternative denominations) is placed both by ISCO-08 and ESCO under the “Science and Engineering Professionals” sub-major group while also recognizing its membership to the health workforce. In Romania, the same occupation is inexplicably moved to the “Health Professionals” sub-major group, without any connection with the international and European classifications. The misplacement of this occupation might be explained, as we said, by a confusion made between the knowledge needed for a profession and the actual place to exert that profession. Indeed, besides research on the one hand, and the industrial sector that attracts a large amount of biomedical engineers on the other hand, the third other main field of work is within the healthcare institutions.

We concluded the last section by arguing that Romania mainly educates for the medical engineering specialization, both at B.Sc. and M.Sc. level, a category that is placed under the engineering sciences label (as well as the one of bioengineer, which is also present in the governmental documents). In this section, we observed that, at the national level, under the occupational aspect, the medical bioengineer is assimilated to the health specialists, despite the fact that the internal educational records, as well as the European and international classification, view this occupation as a part of the engineer professionals.

## 4. Conclusions

We have indicated in this article that a taken-for-granted unquestioned mimetic adhesion to European and/or international classification schemes could sometimes affect the national and/or, respectively, the European attempts to improve the existing qualifications and professional recognition frameworks. In our view, for the BME sector, there are several measures to be taken at the international and EU level in order to speed up the institutionalization of this field. Subsequently, at the Romanian study case level, there should be additional efforts to alleviate the poor operationalization of the BME qualification and profession as a result of the poor national administrative capabilities of properly implementing the aforementioned schemes. In this respect, we present several recommendations and policy proposals.

At the international level, a main action point envisages strengthening the BME status by refining its profile and by creating a specific independent category in the ISCO-08 classification system within the engineering professionals cluster [11]. The description of this specialized unit group of engineering could follow the WHO proposal referring to “engineering professionals that apply knowledge of engineering and medical field to health-care systems to optimize and promote safer, higher quality, effective, affordable, accessible, appropriate, available, and socially acceptable health technology to populations” [11] (Annex 7, pp. 218–219), a clean perspective of this inter-disciplinary field. After a comprehensive listing of the tasks associated with this profession, the WHO input is worth mentioning for three other aspects: first, for presenting several examples of occupations that can be classified as “biomedical engineer”: biomedical engineer as such, but also electro-medical engineer, clinical engineer and medical engineer; second, for preserving the disclaimer that assures the connection of the BME professionals to the health workforce; third, for continuing the efforts of improving the ISCO-08 classification by proposing a similar approach to a proper independent definition of the “biomedical engineering technician”, currently assimilated to a unit group of “Physical and engineering science technicians not elsewhere classified”.

At the European Union level, a similar pattern of updating ESCO with a proper place for the BME professionals should also be undertaken. One should also not forget that the European Union made efforts in order to assure a minimum compatibility or, at least, understanding, among the diverse domestic qualifications schemes, and, since 2008, developed the European Qualification Framework (EQF) as a distinct tool aimed to contribute to the recognition of the qualifications process. In spite of the success that EQF and other internal (Directive 2005/36/EC [37]; Council 2018/C 444/01 Recommendation [38]) or external (the 1997 Convention on the Recognition of Qualifications concerning Higher Education in the European Region [39]; the framework of qualifications of the European Higher Education Area—adopted in 2005 within the larger Bologna Process [40]) instruments had, there is room for progress regarding the development and reviewing of (inter)national qualification frameworks, the focus on learning outcomes, the connections with the labor market and other stakeholder needs, the support for vocational education and training, the validation of non-formal and informal learning, etc. ([41]; see, for example, the demand on registering and regulating across Europe the medical physics experts described in [42]). In this respect, considering the subject of our article, we argue that BME should also be included in the EU Professional Qualifications Directive [37].

Also at the EU level, there should be concrete initiatives to enhance the competitiveness of the EU biomedical engineering sector by not only (a) supporting a unitary supranational vision on the area, in terms of the type of qualifications offered, professional standards created or in terms of policy frameworks aimed at regulating the European market for medical devices, but also by (b) increasing the visibility of the biomedical engineering and its economic role/input and by (c) encouraging new research directions, initiatives, transfer of best practices and strategies after wider and more in-depth consultations between various European and national stakeholders of the biomedical engineering sector (inter/national authorities, industry, regulatory agencies, universities, patients, relatives, medical staff, etc.) [14,43]. Indeed, there are numerous challenges to confront, despite an increase in the number of specializations and students: a lack of common shared policy vision, both in aspects of education and certification, budgetary constraints that usually affect research and teaching in terms of staff, logistics, etc. [44]. Meanwhile, a joint EU approach is an opportunity that would improve student and academic staff mobility and would strengthen inter-university collaboration in designing joint BME programs or sharing best practices.

At the (Romanian) national level, there is an urgent need for connecting the national classification of occupation with the European and the international one. In what concerns the BME field, this means to at least accurately place the biomedical engineers under the engineering professionals minor and unit groups. Of course, Romania could take the lead by following the WHO proposal of defining a singular independent unit group of biomedical engineers. Moreover, considering that Romania mainly educates for the medical engineer qualification, which is not currently connected to COR, the proper identification of the BME field should also reflect all of the occupations that can be gathered under the BME umbrella: biomedical engineer, but also medical engineer, etc. The input of a professional association should also count here, summing up our argument: “Biomedical Engineering is a synonymous of Medical and Biological Engineering. The latter is more accurate, and should be preferred in formal contexts” [12]. 

A second national level policy proposal/recommendation envisages increasing the employability at national level through the opening of the internal labor market for BME specialists in order to operate in hospitals or in health-care services, as their input is needed in various activities, such as testing laboratories, accreditation laboratories, public procurement commissions (for medical devices and logistics, hospital furniture, etc.), the design of medical devices (implants, prostheses, consumables and instruments) and training for handling equipment.

In the end, referring to the research hypotheses of this article, we argue that the analysis of the collected data validated both of them, confirming the existence of a deficient operationalization of the BME qualification and profession, especially at the Romanian national level, and indicating that the existing domestic shortcomings are both a consequence of an unquestioned process of adopting European and even international classification schemes, and of insufficiently developed national administrative capabilities to properly implement the aforementioned classification frameworks.

## Figures and Tables

**Figure 1 ijerph-19-09004-f001:**
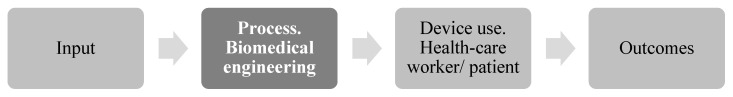
Medical devices process from policies to health outcomes. Adaptation from [11] (p. 20).

**Table 1 ijerph-19-09004-t001:** Authors’ vision on the multiple disciplinarity approaches.

Cross-Disciplinary	Multi-Disciplinary	Inter-Disciplinary	Trans-Disciplinary
It involves distinct input from other disciplines (knowledge, methodology, practices) in the teaching/learning/researching activities of a discipline, but without integrating their content [21], while subsuming to the internal logic of that main discipline	Rather close to cross-disciplinary, it focuses on specific problems that are tackled (simultaneously or consecutively) from the perspectives of various disciplines without any trespassing of their borders in a limited collaborative framework; despite using separate methodologies, each discipline can benefit from the know-how developed in the counter-part disciplines; “the outcome is the sum of the individual parts” [20,22,23]	It assumes a crossing of the borders of different disciplines, either with a transfer of methodologies or with a synthetic coordination of their input “into a coordinated and coherent whole [with] new common methodologies, perspectives, knowledge” [20,22]. This approach sometimes leads to new integrated research areas, but the scientists tend to be influenced by their original disciplinary formation [23]; “the outcome is more than the sum of the individual parts” [20]	It implies a concrete melting of different disciplines (as well as non-scientific input from a larger class of stakeholders) within a distinct new unitary disciplinary framework [20], between, across and beyond the existing knowledge [22]
Less collaboration 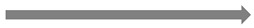 More collaboration

**Table 2 ijerph-19-09004-t002:** Connection between broad educational area and BME specializations in Romania, 2021–2022.

Broad Educational Area	Specific Branch of Science/Educational Area	Field of Doctoral/Master’s Degree Studies	Field of Bachelor’s Degree Studies	Specialization	No. of Credits
Engineering Sciences	Mechanical engineering, mechatronics, industrial engineering and management	Applied engineering sciences	Applied engineering sciences	Medical engineering	240
				Bioengineering	240

Source: [29].

**Table 3 ijerph-19-09004-t003:** B.Sc. level BME specializations in Romania, 2021–2022.

No	University	Faculty	Field of Bachelor’s Degree Studies	Specialization/Bachelor’s Degree Studies Program	Accreditation (A)/Provisional Operating Authorization (PA)	Number of Transferable Study Credits	Maximum Number of Students Who Can Be Enrolled
1	University “Politehnica” of Bucharest	Faculty of Materials Science and Engineering	Applied engineering sciences	Medical engineering	A	240	60
2		Faculty of Medical Engineering	Applied engineering sciences	Medical engineering	A	240	0 * * Specializations/bachelor’s degree study programs for which no admission is organized in the academic year 2021–2022.
3	University “Transilvania” of Brașov	Faculty of Product and Environmental Design	Applied engineering sciences	Medical engineering	A	240	60
4	Technical University of Cluj Napoca	Faculty of Electrical Engineering	Applied engineering sciences	Medical engineering	A	240	75
			Applied engineering sciences	Medical engineering (in Bistrita)	A	240	50
5	University “Dunărea de Jos” of Galați	Faculty of Engineering	Applied engineering sciences	Medical engineering	PA	240	60
6	University “Grigore T. Popa” of Medicine and Pharmacy in Iași	Faculty of Medical Bioengineering	Applied engineering sciences	Bioengineering	A	240	90
7	University “Constantin Brâncuși” of Târgu Jiu	Faculty of Engineering	Applied engineering sciences	Medical engineering	PA	240	30
8	University “George Emil Palade” of Medicine, Pharmacy, Science and Technology from Târgu Mureș	Faculty of Engineering and Information Technology	Applied engineering sciences	Medical engineering	PA	240	60
9	Politechnica University of Timișoara	Faculty of Mechanics	Applied engineering sciences	Medical engineering	A	240	45

Source: [29].

**Table 4 ijerph-19-09004-t004:** B.Sc.-level BME specializations in Romania, 2021–2022.

	Field of Master’s Degree Studies	Name of Master’s Degree Program	Maximum Number of Students Who Can Be Enrolled
University “Politehnica” of Bucharest	Applied engineering sciences	Medical engineering ** ** Master’s degree programs from the structure of the higher education institution included by ARACIS in the category of research master	450 per field

Source: [28].

**Table 5 ijerph-19-09004-t005:** Author’s comparative analysis of the ISCO-08, ESCO and COR of the biomedical engineer occupation.

Register	Major Group	Sub-Major Group	Minor Group	Unit Group	Occupation	Observations
*International level* ISCO 08-International Standard Classification of Occupations [30]	2 Professionals	21 Science and Engineering Professionals	214 Engineering Professionals (excluding Electrotechnology)	2149 Engineering Professionals Not Elsewhere Classified	Biomedical engineer	“It should be noted that, while they are appropriately classified in this unit group with other engineering professionals, biomedical engineers are considered to be an integral part of the health workforce alongside those occupations classified in Sub-major Group 22: Health Professionals, and others classified in a number of other unit groups in Major Group 2: Professionals” ([30], p. 120).
		22 Health Professionals	226 Other Health Professionals	2269 Health Professionals Not Elsewhere Classified	-	“In using ISCO in applications that seek to identify, describe or measure the health workforce, it should be noted that a number of professions considered to be a part of the health workforce are classified in groups other than Sub-major Group 22: Health Professionals. Such occupations include but are not restricted to: addictions counsellors, biomedical engineers, clinical psychologists and medical physicists” ([30], p. 125).
*European level* ESCO—European Skills, Competences, Qualifications and Occupations [31]	2 Professionals	21 Science and Engineering Professionals	214 Engineering Professionals (excluding Electrotechnology)	2149 Engineering Professionals Not Elsewhere Classified	2149.4 Biomedical engineer	*Overlapping ISCO 08 classification and denomination for the major, sub-major, minor and unit group, as well as for the occupation.* “It should be noted that, while they are appropriately classified in this unit group with other engineering professionals, biomedical engineers are considered to be an integral part of the health workforce alongside those occupations classified in Sub-major Group 22: Health Professionals, and others classified in a number of other unit groups in Major Group 2: Professionals” [31].
		22 Health Professionals	226 Other Health Professionals	2269 Health professionals not elsewhere classified	-	“In using ISCO in applications that seek to identify, describe or measure the health workforce, it should be noted that a number of professions considered to be a part of the health workforce are classified in groups other than Sub-major Group 22: Health Professionals. Such occupations include but are not restricted to: addictions counsellors, biomedical engineers, clinical psychologists and medical physicists” [31].
*National level* COR 2021 Classification of occupations in Romania [35,36]	2—Specialists in various fields of activity	21—Specialists in the field of science and engineering	214 Engineers (excluding Electrotechnology)	2149 Engineers and assimilated [occupations] unclassified in previous unit groups	No listed occupation	*Overlapping ISCO 08 classification and denomination for the major, sub-major, minor and unit group, but not also occupation.*
		22—Health specialists	226 Other health care specialists	2269 Specialists in the field of health not classified in the previous unit groups	226,904 Medical Bioengineer	*Overlapping ISCO 08 classification and denomination for the major, sub-major, minor and unit group.* *However, the occupation is not foreseen in this unit group by ISCO-08 or by ESCO.*

## Data Availability

Not applicable.

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
