# Peer review of "Regulation and Certification of (Bio)Medical Engineers: A Case Study on Romania"

_ijerph, 2022, doi:10.3390/ijerph19159004_

Round 1

Reviewer 1 Report

The abstract is clear and covers all aspects needed.

I suggest to introduce “Romania” among the key words.

In the chapter “Materials and Methods” it is missing a clear presentation of the methodology of the research. Questions such as: What methods have been deployed? How was the data collected and analyzed? What are the research hypotheses?  What is the main research objective of the article? have to addressed in an academic paper.

The ideas depicted in the Chapter “BME’s Societal Role” do not tackle the topic of “Materials and Methods” (which should be presented in the paper), they just emphasize the importance of BMEs.

In my opinion the authors need to emphasize the research and novelty character of the present paper. The literature review research should rely also on scientific papers from the last years. The scientific article represent only a small share of the bibliography and this aspect should be improved.

It would be useful to know if similar studies have been conducted about the BME field in other countries or about other filed of activity (eventually in Romania).

The paper is a good country report which reveals an in depth understanding of the issues existing in Romania in the field of BME profession. However, in order to be published, a more scientific approach of the methodology and literature review is needed. It should be also highlighted why this paper is relevant for the academic community in the field and maybe some generalization possibilities of the outcome could be identified.

Author Response

Dear Reviewer,

Thank you very much for your comments and suggestions following the assessment of the article.

After receiving your report, we have revised the manuscript and:

- we have introduced Romania among the keywords;

- we have offered more details about the methodology used, and we clarified the objective and hypotheses of this research;

- we have emphasized the novelty character of the paper, as well as its relevance for the academic community.

We have to underline here that your suggestion regarding additional scientific articles to be mentioned in the literature review section is extremely valuable, but, in the case of the present paper, there are no similar researches within the last years. In fact, this literature state-of-the-art represents arguments for the novelty and relevance of the paper. However, in this revised form, we have included references to several recent contributions that, by far, are connected to the subject of this article.

Best regards,

The authors

Reviewer 2 Report

The theoretical part of the paper presents a relevant issue which gave arise to a new question to be addressed to literature. Introduction is well described and arguments are coherent and sustained with works published by specialists (entities and authors). Methodological part is coherent with the questions raised during the introduction. And, Conclusions appears in line with the previous parts, especially in which concerns with the questions raised.

However, i suggest: 1) the paper is focused on BME and i think it could be important to compare or bring some information about other cases in Romanian or European contexts, that already are successfully conceptualized and implemented; 2) some statements should be more cautious, as an example in the end of Conclusions: “In the end, we argue that the research question of this article revealed the existing of a deficient operationalization of the BME qualification and profession, especially at the Romanian national level, and that the existing domestic shortcomings are both a consequence of an unquestioned process of adopting European and even international classification  schemes,  and  of  insufficiently  developed  national  administrative  capabilities  to  properly implement the before mentioned classification frameworks”.

Author Response

Dear Reviewer,

Thank you very much for your comments and suggestions following the assessment of the article.

After receiving your report, we have revised the manuscript and we have tried to express in a more cautious manner the conclusions of the paper, in connection with the research hypotheses.

We have to underline here that your suggestion regarding additional scientific articles to be mentioned in the literature review section is extremely valuable, but, in the case of the present paper, there are no similar researches in Romania and in European contexts within the last years. In fact, this literature state-of-the-art represents arguments for the novelty and relevance of the paper. However, in this revised form, we have included references to several recent contributions that, by far, are connected to the subject of this article.

Best regards,

The authors